

# Quantum wavefront shaping with a 48-element programmable phase plate for electrons

Chu-Ping Yu[1,2], Francisco Vega Ibañez[1,2], Armand Béché[3] and Johan Verbeeck[1,2*]

**1** University of Antwerp, EMAT, Groenenborgerlaan 171, 2020, Antwerp, Belgium
**2** NANOlab Center of Excellence, University of Antwerp, Groenenborgerlaan 171, 2020 Antwerp, Belgium
**3** AdaptEM, Interleuvenlaan 62, 3001 Heverlee, Belgium

⋆ jo.verbeeck@uantwerpen.be

## Abstract

We present a 48-element programmable phase plate for coherent electron waves produced by a combination of photolithography and focused ion beam. This brings the highly successful concept of wavefront shaping from light optics into the realm of electron optics and provides an important new degree of freedom to prepare electron quantum states. The phase plate chip is mounted on an aperture rod placed in the C2 plane of a transmission electron microscope operating in the 100-300 kV range. The phase plate's behavior is characterized by a Gerchberg-Saxton algorithm, showing a phase sensitivity of 0.075 rad/mV at 300 kV, with a phase resolution of approximately $3 \cdot 10^{-3}\ \pi$. In addition, we provide a brief overview of possible use cases and support it with both simulated and experimental results.


# 1 Introduction

Wavefront shaping, or the spatial and time-dependent control over the phase in coherent waves, has revolutionized many diverse scientific fields ranging from radio and light astronomy [1, 2], radar [3], acoustics [4–6], seismology [7], telecommunication [8, 9] and many more [10]. It requires a device that can apply a position-dependent phase change and can be augmented by adding a control loop to obtain adaptive optimization of the wave with respect to some goal function. In optics, this can be realized with so-called spatial light modulators, which can be based on moving arrays of mirrors or by liquid crystal-based setups that can change refractive index when applying an electric field [11, 12].

Matter waves, as introduced by de Broglie [13], are also amenable to this same concept. Indeed, the working principle of an electron microscope is entirely based on describing the free electrons as coherent quantum waves with wavelengths of the order of picometers. The capability of manipulating these electron waves is an indispensable part of a transmission electron microscope (TEM). The most relevant addition to phase manipulating devices in recent decades is, without a doubt, the spherical aberration corrector [14–16], which flattens the phase front of the electron wave induced by (high-order) geometric aberrations of the microscope lenses and allows the forming of a sharper and more intense probe in scanning probe applications. Removing unwanted phase aberrations has significantly increased the resolution and current density of the scanning transmission electron microscope (STEM) with many benefits in, e.g., spectroscopic applications.

Besides canceling geometric aberrations, the ability to arbitrarily shape the electron wavefront is gradually gaining attention with the hope of improving contrast or selectivity in electron microscopy setups. There has been a renewed surge of such phase modulators and their applications in the past few years. In soft material imaging, different phase plates such as Zernike [17, 18], Boersch [19, 20], Zach [21, 22], or Volta [23–25] have been implemented in the TEM to imprint a constant phase shift to a (central) part of the electron wave, to increase the contrast when imaging phase objects. Some other designs with relatively higher complexity may modify both the amplitude or phase configuration of the electron wave to create an electron probe of specific shape [26], to increase contrast [27, 28], or to extract specific information from the electron-sample interaction [29, 30], to name a few. Some of these complex modulators even exhibit control over the parameters or magnitude of the modulation. The electrostatic phase plate reported by Verbeeck et al. [31] has demonstrated changes in interference between 4 partial waves by altering their mutual phase relation. Barwick and Batelaan [32] showed that a pulsed laser beam could induce a phase shift in the electron beam and that the contrast of the formed image can be optimized by tuning these laser pulses. Different realizations of using the ponderomotive force to change the phase of an electron beam appeared [33–36]. The electrostatic phase plate reported by Tavabi et al. [37] has demonstrated a tuneable azimuthal phase by setting up specific electric field boundary conditions, which was interpreted as adding orbital angular momentum to the electron beam.

Here we report on an adaptive electrostatic phase plate based on the proof of principle demonstration by Verbeeck et al. [31], but with significantly increased complexity, performance, and practical usefulness. The phase plate consists of 48 openings, or pixels, transparent to an incoming coherent electron wave. The vertical walls of the pixels are made into electrodes so that an electric potential can be established inside, changing the wavelength of that part of the transmitted wave. Since separate voltage sources control each of the 48 pixels, the phase of the entire transmitting coherent electron wave can be programmed at will. This design and the electrostatic nature grant the phase plate several advantages, such as short response time, the ability to realize complex and arbitrary phase configurations, low power dissipation, compactness, low weight, and high stability and repeatability.

The experimental part of the paper provides a concise summary of the reported phase plate. The design of the phase plate is described first, as well as the components and mechanism to create a phase shift on an electron wavelet. The manufacturing design choices are briefly discussed in the scope of the challenges faced. The device's optical performance is then evaluated regarding its phase sensitivity and response time.

We discuss the applications of the phase plate in the scope of electron microscopy. Using the unique properties of a fast, hysteresis-free programmable phase plate, we demonstrate how novel imaging setups can expand or improve imaging modalities in TEM. We provide simulated examples and early experimental attempts towards electron wave modulation, complex sampling schemes, adaptive optics, and phase-coded ptychography to hint at what phase plates could bring to the electron microscopy community.

## 2 Experimental considerations

### 2.1 Description of the electrostatic phase plate

The basic working principle of the phase plate is sketched in Figure 1-a. A coherent incoming electron wave is made to interact with an insulating membrane that has several holes. The top and bottom surfaces of the membrane are covered with a ground shield, while the inside of the holes is coated with a conductive layer that can be put to a controlled electrostatic potential ($V_1$ and $V_2$ in the simplified sketch). The potential surrounding the holes creates a potential landscape for the fast electrons that accelerates the electron upon entering and decelerates upon leaving this area. This will cause a phase change between the partial waves leaving these holes where one could imagine them as coherent Huygens sources that will constitute a now phase-programmed wave upon propagation in free space. The phase shift $\phi$ obtained is given by the electrostatic Aharanov-Bohm shift:

$$\phi = \frac{\pi e}{\lambda E_0} \int_\Gamma V(\vec{r})dl \, . \tag{1}$$

For an electron wave with wavelength $\lambda$ and energy $E_0$ and crossing a region of space with an electrostatic potential $V(\vec{r})$ along a trajectory $\Gamma$. In the case of a weak perturbation, the electron's trajectory is not altered by this field, and the phase shift becomes directly related to the projected electrostatic potential. The goal of a pixelated phase plate is to create a potential profile that, in projection, leads to a constant phase shift proportional to the voltage applied to each pixel element. This occurs if the projected potential changes as little as possible over the region of each hole, which can be obtained by choosing a high aspect ratio (height/diameter> 1).

From a practical perspective, the AdaptEM WaveCrafter phase plate [38] comprises three main elements shown in Figure 1b-e: a dedicated condenser aperture holder containing the phase plate chip, a 48-channel programmable voltage source and a remote computer for control and user interface, respectively. The phase plate used in this work is composed of 48 independent active elements, or pixels, arranged in 4 concentric rings and 12 petals (see Figure 1b). Each element consists of a layered structure similar to the one described by Matsumoto and Tonomura for a single phase shifting element [39]. An aspect ratio of approximately 2 was chosen to avoid lensing, and a total diameter of the active area of 50 $\mu$m assures that a modern electron microscope can coherently illuminate the whole device.

One considerable advantage of this phase plate design lies in the relatively low voltage (in the mV range) required to induce a phase shift of $2\pi$. This avoids high electric field breakdown issues in the nanoscale features of the chip and has the benefit that readily available voltage sources, which are simultaneously precise, stable, low power, fast and reliable, can be used.

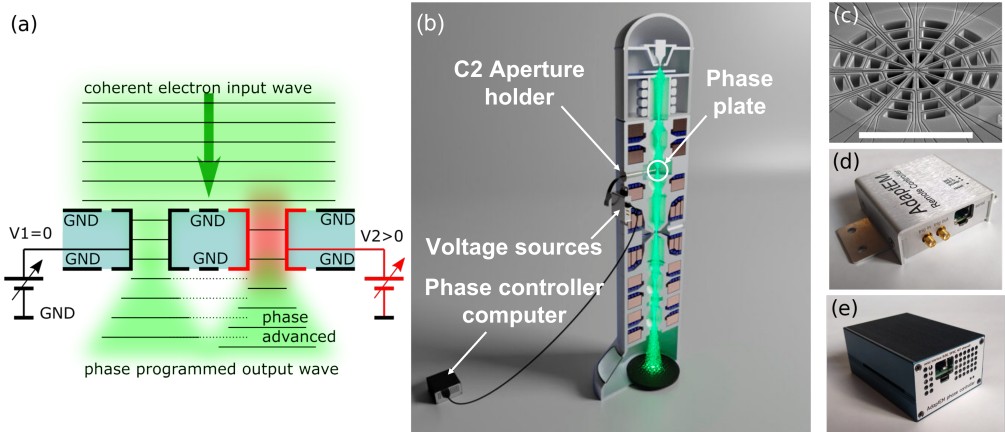

Figure 1: Sketch of the working principle of the phase plate (a). Only 2 pixels are drawn. 3D render of the setup (b) and the main components, including the phase plate (c), the voltage sources (d), and the phase controller computer (e). A reference bar of 30 $\mu$m is presented in (c).

## 2.2 Characterization

To experimentally examine the projected potential profile, phase reconstruction based on the Gerchberg-Saxton (GS) algorithm [40] was performed on a set of images of the phase plate, where each pixel is excited with increasing electrostatic potential (48 pixels, 11 voltage levels, 528 images in total). For the characterization, the phase plate is inserted in the sample plane of an FEI Tecnai Osiris S/TEM operating at 200 kV and illuminated with a parallel electron beam. The images are taken from the back focal plane of the objective lens (diffraction mode), while the objective lens is largely defocused so that the detector can capture the near-field diffraction pattern of the phase plate. This experiment is aimed to characterize the projected potential on the phase plate when varying the phase inside each pixel in a range between 0 and $2\pi$. A rough estimation of the voltage corresponding to a $2\pi$ phase shift was first found by assigning a gradually increasing voltage to half of the pixels randomly and repeatedly. Theoretically, a $2\pi$ phase shift should not result in any difference in the diffraction pattern formed by the phase plate. Thus a visual inspection of the voltage at which the pattern shows the least variation over time is a reasonable estimation of the value at which the pixels yield a $2\pi$ phase shift. Once this voltage $V_{2\pi}$ was found, a series of images with 11 different potentials equally spread between 0 and $V_{2\pi}$ was taken for each pixel.

The defocused condition was specifically chosen so that outgoing waves from the electrodes interfered strongly with each other, and the phase difference between separate neighboring wavelets is significantly encoded in the recorded intensity images (see supplementary). This choice of detection plane was preferred over recording at an in-focus condition that interferes all of the wavelets together for several reasons. First of all, at the right focus, the transmitted electrons are concentrated in a very small region (less than 1 % of the size of the recorded defocused images), and creating a high enough camera length to sufficiently sample such patterns on a pixelated camera for phase retrieval is not trivial. On top of that, the inversion invariant nature of the wave intensity in the reciprocal space would also challenge obtaining a unique reconstruction and greatly hinder the retrieval algorithm's convergence.

The result of the reconstruction is summarized in Figure 2. The phase response of all pixels, as they were individually excited, is fitted using a linear function, representing the phase sensitivity of that pixel to the applied voltage. A phase sensitivity matrix can be constructed showing the phase sensitivity of pixel $i$ upon exciting pixel $j$. The phase sensitivity matrix in

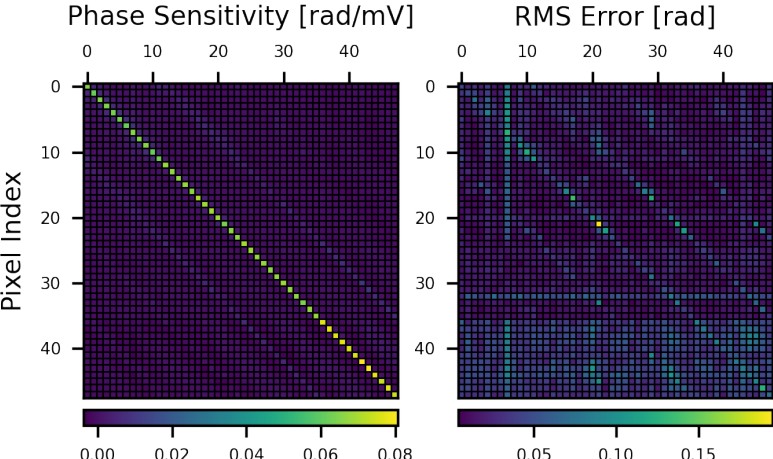

Figure 2: Phase sensitivity matrix and the corresponding root mean squared error of the linear fitting.

Figure 2 shows a strong response on the diagonal, meaning that the excited pixel is the only one showing a significant linear phase shift against the applied voltage. An average phase sensitivity of 0.075 rad/mV is found, which translates to a theoretical phase resolution of approximately $3 \cdot 10^{-3}$ $\pi$ according to the smallest step size provided by an ideal 16-bit DAC (maximum 2.5 V, smallest step $2.5 \times 2^{-16}$ V). The error matrix, also shown in Figure 2, indicates response deviation from the expected linear behavior, mainly resulting from imperfections in the phase retrieval process, such as the finite pixel size and non-ideal detector response. These can cause a difference between the recorded intensity and the actual waveform. The error is calculated by the root mean square error of the fitted result, which is found, at maximum, to be 3% of $2\pi$ (0.19 rad), while on average less than 0.5% of $2\pi$ (0.027 rad).

Besides the expected response of the phase plate, it is equally important to characterize any non-ideal behavior. The inhomogeneity describes the phase deviation within the pixel area from the ideal constant, homogeneous expectation. We evaluate the standard deviation of the reconstructed phase within each activated pixel and find it to be < 1.7% of $2\pi$. The cross-talk refers to the phase response within a pixel region caused by the voltage applied to another pixel. We estimate this as the maximum linear response of a non-excited pixel as a function of any other excited pixel. The off-diagonal lines found exactly 12 pixels away from the main diagonal in both matrices in Figure 2 indicate that the strongest cross-talk is, unsurprisingly, found between neighboring pixels due to how the pixels are ordered in the matrix (see supplementary). The cross-talk is measured to be < 0.012 rad/mV, which amounts to 15% of the response of the excited pixel. In summary, the inhomogeneity only creates phase error much less than $\frac{2\pi}{10}$, which is generally accepted as very good in light optics [41,42], while the cross-talk is clearly the biggest contributor to a non-ideal response. This behavior could be significantly improved in the next design iteration, where an additional top-ground layer could shield the effects from neighboring pixels and the conductive tracks leading to those pixels.

Characterizing the temporal response of the phase plate is also important for applications that rely on rapid switching between different electron probe shapes or phase configurations. Since the phase shift results from the projected potential in the electrodes, the response of the phase plate can be characterized by the time required to build up the potential. With the criterion of phase error < $\frac{2\pi}{10}$, the response time is measured to be less than 1.3 $\mu s$ for reaching from 10 % to 90 % of $V_{2\pi}$ and is entirely dominated by electronics.

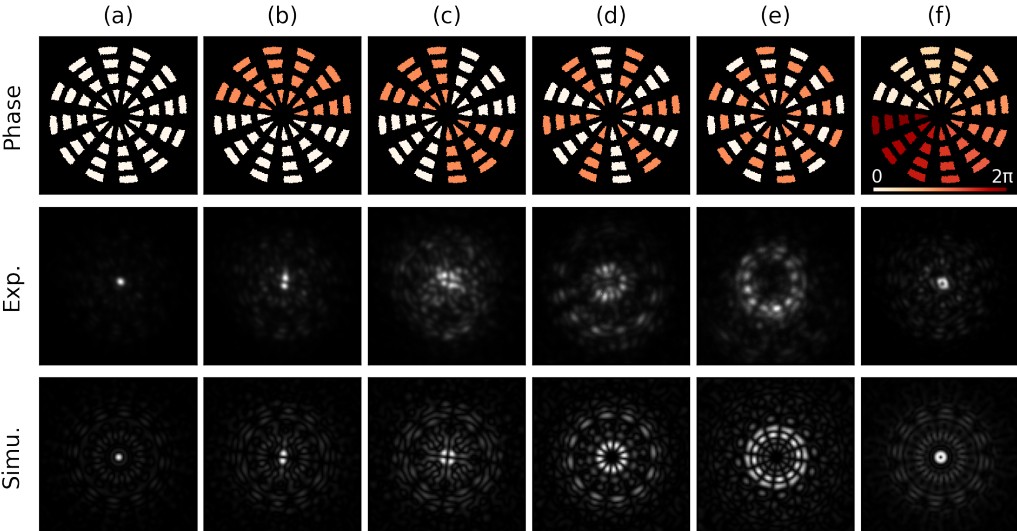

Figure 3: Realization of various electron quantum states. The three rows of figures, from top to bottom, are the phase configurations set on the phase plate, the simulated probe shapes, and the resulting experimental probe images, respectively. Note the excellent agreement between expected and obtained results showing successful arbitrary wavefront shaping.

# 3 Application examples

## 3.1 Designer electron waveforms

To demonstrate the capability and visualize the effects of a freely programmable phase plate, we recorded the far-field diffraction patterns of various phase-modulated electron waves in a TEM (Figure 3). These patterns form rather complex configurations compared to ones formed by commonly-used round apertures, even when all phase plate elements are at ground potential. This is due to the amplitude modulation created by the set of holes, which produces highly delocalized tails. Previous theoretical research points out that the proportion of the electrons in these tails is directly related to the fill factor (% of the electron wave not blocked by the material of the phase plate) of the probe forming aperture [43]. Although improvement has been made on the fill factor (current design approximates 30%, while the proof of concept 2x2 version from 2018 [31] had only 17%), a large proportion of the electrons can still be expected in the tails.

A close comparison between experimental intensity profiles and simulations is found. From Figure 3, columns (b-e) show a phase shift of $\pi$ applied to half of the total pixels with different patterns; therefore, the original single intense spot in the diffraction pattern is split into multiple parts due to destructive interference. Double-spots (b), quadruple-spots (c), and even a duodecuple-spot (d) consisting of six $0 - \pi$ pairs are shown. By taking into account the radial distribution of the rings, a checkerboard-like pattern (e) can be created. These patterns cover a few of the 48-dimensional Hadamard basis [44], which defines an orthogonal basis consisting entirely of pixels with either 0 or $\pi$ phase. Lastly, (f) shows the result of a vortex setup with an orbital angular moment equal to 1 [45]. This is done by creating a phase ramp from 0 to $2\pi$ in the azimuthal direction. The vortex can be verified by the signature singularity point at the center of the resulting probe approximating one member of the Laguerre-Gaussian orthogonal basis set [46].

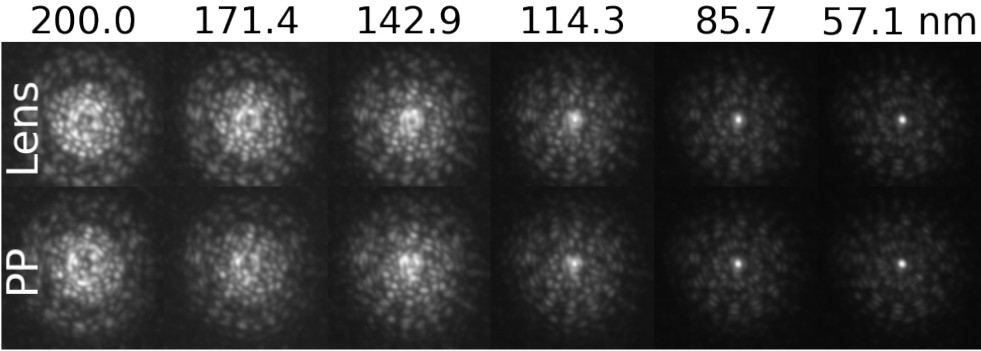

Figure 4: Defocused probes formed by defocusing the microscope lenses (top row) and the phase plate (bottom row) at 300 kV acceleration voltage and an opening angle of 1 mrad. Note the close similarity of both, showing that the phase plate can mimic the action of a round lens to up to 200 nm defocus.

The phase plate can also create a phase profile imitating geometric optical elements and aberrations. Typically they can be modeled by a phase shift that follows a Zernike polynomial in the angle with respect to the optical axis [47]. How faithfully the phase plate can recreate such polynomials at different angles has been discussed theoretically in detail by Vega Ibáñez et al. [43] and relates to parameters such as the order of aberration, the fill factor, number of pixels, and pixel shape. Here, a defocus effect (second-order in angle) is introduced by either the conventional electromagnetic objective lens of the microscope or by the phase plate to demonstrate this concept. The resulting probe shapes are shown in Figure 4, respectively. The two rows show good resemblance with each other up to 200 nm defocus. Further defocusing causes a steep phase ramp within the area of the individual pixels, which can not be faithfully reproduced anymore by the phase plate. For this reason, the phase plate can obviously not replace an actual (round) lens of any significant strength.

## 3.2 Object sampling with different wavefunctions

Electron microscopy is a process of sampling an unknown material with an electron wave. Once the incident wave interacts with the examined object, the information from the object is imprinted on the wave by creating changes in amplitude, phase, and the creation of inelastic scattering signals. When the measurement result of the interaction between the object and a beam with a given electron waveform provides insufficient information about the sample, different waves can be used to interrogate the object. For example, in-line holography [48–50] is done by recording the intensity of a beam while varying the phase by changing the defocus of the objective lens. STEM essentially describes a process to accumulate information about the material by a dense sampling while spatially scanning a localized electron beam. In both cases, multiple measurements while changing the incoming electron wave enriches the acquired information and eliminates confusion that can sometimes not be resolved with a measurement process that only uses a beam with a single static waveform.

Such multi-waveform sampling schemes rely entirely on the ability to alter the wavefunction of the beam electron states. Even though some form of modulation of the wavefunction is present in any electron microscope (e.g., defocus, beam tilt, beam shift, or aberration correctors), they often rely on electromagnetic elements, which can suffer from slow settling times and hysteresis effects. For example, in the acquisition of through focal series images, an update rate in the order of seconds to minutes is typically applied to induce small focal changes in the objective lens [51, 52].

The phase plate presented here can update to an entirely new pattern in a few $\mu s$ without hysteresis so that complex sampling schemes can be realized efficiently. For instance, the phase plate can cycle over a few different wavefront settings for each probe position in a STEM recording. Compared to through focal TEM acquisition, where the focus is changed between recording image frames, we could now update multiple focus levels for each probe position in a STEM scan, providing, e.g., increased depth of field. This dramatically reduces the difficulty of realigning each image, especially in cases of severe sample drift, and also avoids inconsistencies caused by contamination building up on the sample over time.

Changing defocus is just one of the possible wavefunctions to sample an object of interest. Being a non-orthogonal change to the beam wavefunction, it could be argued that this is not even an optimal choice of basis. The adaptability and rapid response of the phase plate can be extended to a wide variety of orthogonal basis sets that can be specifically chosen to efficiently encode selected knowledge about the sampled object into the probing electron waves.

This concept is widely used in light microscopy and serves as an important cornerstone for techniques such as stimulated emission depletion (STED) microscopy [53–55] and switching laser mode microscopy (SLAM) [56, 57]. Two or more waveforms sequentially illuminate the sample, and the sharp feature created by the difference between the illuminating waves can be exploited to increase the resolution of the final image.

The same concept can now be applied to electron microscopy with a two-fold beneficial effect. Indeed, changing between a probe state with and without orbital angular momentum will slightly improve image resolution due to differential imaging with both probes (super-resolution). But more importantly, this method also cancels the long probe tails arising from the amplitude modulation of the pixel shapes, as these tails are nearly identical for both probe wavefunctions. This is a far more critical effect as it dramatically increases the practical resolution that can be obtained even when the fill factor of the phase plate is not ideal and shows a way to significantly outperform the results presented earlier for the single waveform aberration correction prospects of programmable phase plates for electrons [43]. The result of this differential scheme is demonstrated with high-angle annular dark field (HAADF)-STEM simulation (Figure 5). Electron probes, as results of the far-field diffraction of three illuminating wave functions, created by a phase plate with zero phase, vortex phase, and a conventional round aperture, are used to scan a single-layer hexagonal boron nitride, with 200 keV electron beam energy, a spherical aberration $C_3$ of 1.2 mm and operating at Scherzer defocus, in agreement with a typical uncorrected TEM instrument. The convergence angles of the electron probes are set to 9.5 and 11 mrad for the round aperture and the phase plate, respectively. We select a larger opening semi-angle for the phase plate since its capability to correct aberrations yields an optimal imaging condition at 11 mrad. The subtraction of the vortex image from the plain phase plate is then presented as the difference image.

The simulated images are then juxtaposed to illustrate the effect of the tails, and an intensity profile (orange line) is drawn across each image (at the position of the white dashed lines). Both images from the phase plate have non-zero intensity in the vacuum area (the left half of the simulation box) due to the tails' interaction with the crystal. The profile from the image formed with the round aperture shows much faster decay as the intensity distribution of an aberrated Airy probe is more concentrated. The difference image demonstrates good cancellation of this false background, and the intensity profile quickly converges to zero, with small fluctuations due to slight differences between the tail configuration of the two probes. This result shows that the phase plate can indeed provide an excellent tail effect cancellation when alternating between a flat phase and vortex phase probe. The resolution is significantly improved over the non-corrected round aperture at the expense of some signal loss related to the fill factor and the loss of low-frequency sample information. This demonstrates the potential for aberration correction with a device that is significantly smaller (< 5 mm), lighter, faster

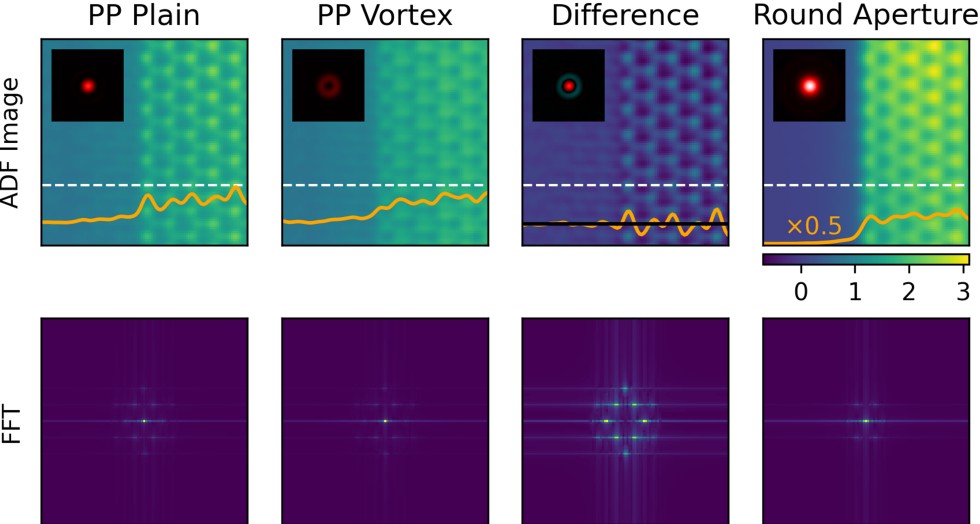

Figure 5: Simulated ADF images of various probe shapes (see the insets) and their Fourier transforms. The line profiles (orange lines) are taken at the position of the white dashed line in each image. Note that the intensity profile in the round aperture image is halved for better presentation and that the black line in the difference image indicates zero, while in other images zero is set at the bottom of the figures.

($\mu s$), more energy efficient ($< 5$ W), and requires far less stringent control over the precision of the voltage/current sources as compared to current multipole correctors.

## 3.3 Adaptive optics

Using the fast and hysteresis-free phase programming offered by the electrostatic phase plate opens the attractive possibility of adaptive optics. As a proof of concept, such a setup is realized (fig. 6). An algorithm repeatedly reshapes the electron probe with the phase plate in order to reach a higher variance in the high-angle annular dark field (HAADF) image, which is taken as a figure of merit that links with 'image sharpness' [58].

The algorithm sequentially adds phase modifications from a list of discretized low-order Zernike polynomials to the latest best-performing phase configuration. Zernike polynomi- als are chosen since they exhibit close similarity to common aberrations in the electron microscope and form a complete, orthogonal basis. A HAADF image is consequently recorded with every new probe. If the variance is higher in the new image, the current best is replaced with this new variation. Once all the configurations are tested, their magnitudes in terms of phase value are reduced by half for a further refinement step. The process is demonstrated by inserting the phase plate in the C2 aperture of a probe and image-corrected FEI-Titan operating at 300 kV in microprobe mode with a convergence angle of 1 mrad (to minimize the effect of aberrations and partial coherence effects). The HAADF image is taken from a gold cross-grating test sample with a deliberately intro- duced initial defocus of approximately 1 $\mu$m. The result of the correction is shown in figure 6(c). The process converges after 32 iterations with a sharper resulting image, even though 1 $\mu$m defocus cannot be entirely compensated by the phase plate due to the steep phase profile. The result shows the feasibility of counteracting the lens defocus automati- cally. The process takes approximately 1 minute, but this time is currently dominated by sub-optimal software handshaking between scan engine control, image readout, and phase plate control, and can be dramatically improved in the future. As an estimate, with the assumption that an update can be made by evaluating a minimum area

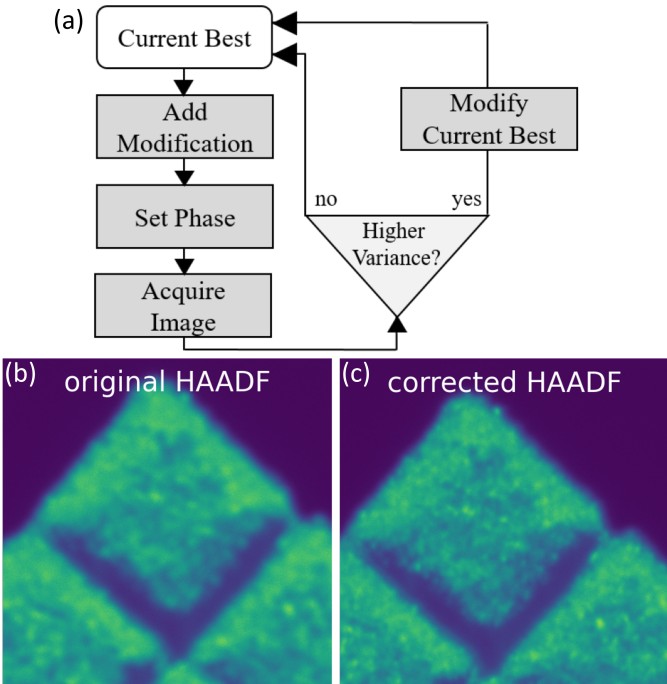

Figure 6: Schematic of the adaptive probe correction with phase plate. The flowchart displayed in (a) demonstrates the fundamental steps of the algorithm being employed. The HAADF images before (b) and after (c) the correction are shown below.

of 100x100 pixels at 1 $\mu$s dwell time (a reasonable dwell time to produce HAADF images with an acceptable noise level), the update rate for the correction scheme would be 1 kHz. This frequency is easily within reach of the phase plate, which currently offers a maximum update rate of 100 kHz, limited by the electronics. This would result in an adaptively optimized image within 10 ms which would be a small fraction of the time to take, e.g., a full 1024x1024 frame. Of course, this time depends on the beam current, as enough image quality is required to make good decisions on the next step. Further work is needed to evaluate the best goal function and most optimum control loop, but the proof of concept demonstrates the scheme's feasibility.

This process could bring significant benefits for the automation of microscopy experiments. Automatic data acquisition and feature identification are widely used for life science research and quality control in the semiconductor industry. With them, the analysis of large amounts of samples can be done without operator intervention, and the demonstrated probe correction scheme can be utilized for maintaining the quality of the optical system over a much longer operation time.

This iterative optimization process can also be extended to any technique in electron microscopy where a specific quantifiable property is related to the shape or phase of the electron probe. For example, in electron energy loss spectroscopy, the intensity of a specific plasmon peak can be tracked while reshaping the electron probe until the optimal probe shape selectively highlights the corresponding plasmon mode [59].

## 3.4 Phase programmed ptychography

Besides shaping a focused electron beam, the phase modulation capability of the phase plate operating under parallel-beam conditions can bring new opportunities for other microscopy applications. For example, coherent diffraction imaging and ptychography can benefit from using the phase plate as a "modulator" or a "diffuser" to break symmetry in the illuminating

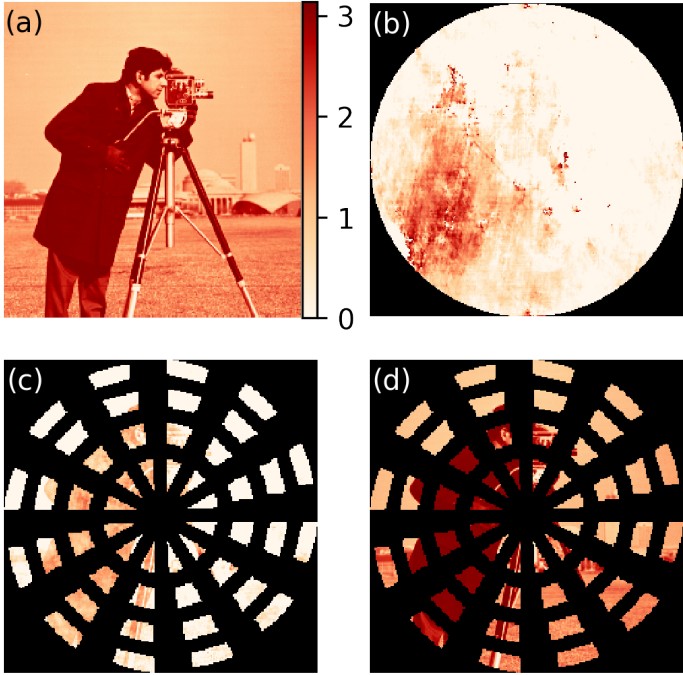

Figure 7: Simulated ptychographic phase reconstruction from recorded diffraction patterns with various illuminating beams. (a) The ground truth phase image of the object. (b-d) Reconstruction results from illuminating beams formed by a conventional round aperture, flat phase plate, and phase plate with random phase configuration, respectively. The dark region indicates the opaque part of the aperture. Note the significant improvement in phase reconstruction quality when the incoming beam is phase randomized. As the object is only illuminated once, reconstruction is only possible in those areas where the amplitude is not zero.

beam and thus increase the robustness and convergence rate of the reconstruction. The benefit of a modulator has been widely reported and studied in the field of light microscopy [60] and electron microscopy [61–63]. Among the reported realizations of ptychography in electron microscopy, with or without a modulator, the reconstruction of the complex object relies on repeated sampling at different locations of the object, with the criterion that the illuminating beam partially overlaps with the sampling at a nearby position. This overlap creates the so-called "information redundancy" [64, 65], which eliminates the twin-image artifact [66] that originates from the central symmetry of the illuminating beam. On the other hand, such symmetry can be easily broken by a random phase configuration introduced by the phase plate instead of the displacement of the beam or the sample.

We hereby demonstrate this concept by performing phase reconstruction on simulated diffraction patterns from a target pure phase object (Figure 7a). The diffraction patterns are generated by different illuminating waves, formed with a round aperture, the phase plate set to zero, and one randomly generated phase configuration. The phase reconstruction is again based on the GS algorithm, and the resolved objects are obtained after 50 iterations. The results are shown in Figure 7(b-d). Neither a round aperture nor a zero phase plate could generate a convincing reconstruction result, as the geometry of both apertures is centrosymmetric. However, introducing a random phase configuration increases the reconstruction quality significantly, despite the sample being only illuminated at one beam position.

The amplitude modulation of the phase plate inevitably results in missing information about the reconstructed object, which could be filled by moving the beam or the sample to

illuminate the whole region of interest at least once. It should be noted here that the phase plate is placed in front of the sample, and all electrons interacting with it are recorded. This means that the limited fill factor does not reduce the electron dose efficiency nor increase beam damage on the sample.

## 4 Conclusion

We report the successful realization of arbitrary wavefront shaping of electrons with a novel 48-pixel programmable electrostatic phase plate. The phase plate is capable of introducing a phase shift of more than $60\pi$, as well as fine-tuning the phase value with step size as small as $3\cdot10^{-3}$ $\pi$ for 300 keV coherent electron beams. Cross-talk between pixels was shown to be < 15% and can be improved further with better shielding electrode geometries. This brings modern adaptive light optics concepts into the domain of electron beam instruments. The rapid response of the device allows up to 100 kHz update rates making it possible to do on-the-fly auto-tuning of differential contrast schemes without a noticeable recording time penalty for the user. The examples demonstrate the potential for a rich field of emerging applications offered by the phase degree of freedom. Immediate use cases focus on electron microscopy, but other electron beam instruments, such as, e.g. e-beam lithography or semiconductor inspection tools, could also profit significantly from this realization. With an even broader perspective, we demonstrate here the arbitrary preparation of coherent quantum states that might be exploited in novel quantum information/computing schemes over a much wider range of electron energies than the ones demonstrated here.

## Acknowledgments

All authors want to thank Gert de Bont for providing the graphics in figure 1 and Stijn Van den Broek for never-ending support with the focused ion beam instrument.

**Funding information**    This project is the result of a long-term effort involving many different sources of funding: JV acknowledges funding from an ERC proof of concept project DLV-789598 ADAPTEM, as well as a University IOF proof of concept project towards launching the AdaptEM spin-off and the eBEAM project, supported by the European Union's Horizon 2020 research and innovation program FETPROACT-EIC-07-2020: emerging paradigms and communities. This project has received funding from the European Union's Horizon 2020 research and innovation program under grant agreement No 823717 – ESTEEM3 and via The IMPRESS project from the HORIZON EUROPE framework program for research and innovation under grant agreement n. 101094299. FV, JV, and AB acknowledge funding from G042820N 'Exploring adaptive optics in transmission electron microscopy.' CPY acknowledges funding from a TOP-BOF project from the University of Antwerp.

## A Defocused images of the phase plate

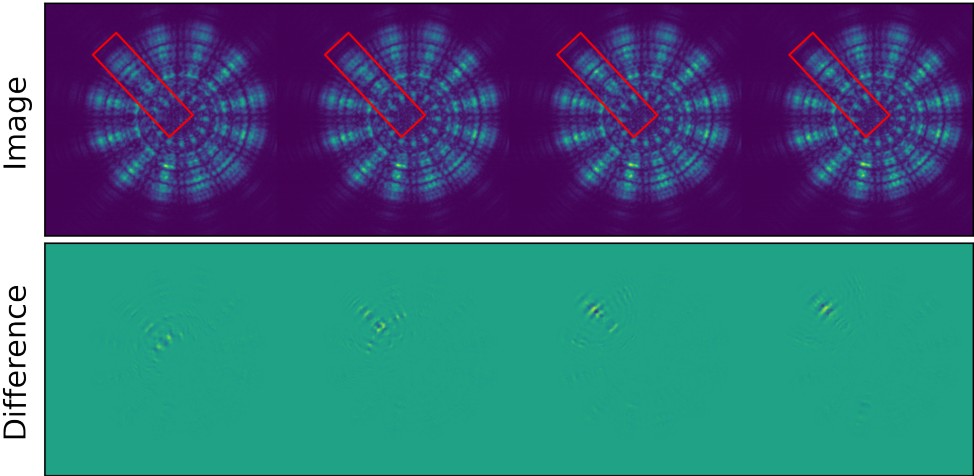

The defocused images of the phase plate with a voltage applied to one pixel in each ring and the difference map between images with and without excited pixel. The changes in the interference patterns imply a local phase shift which is revealed by the phase reconstruction algorithm.

## B Pixel index

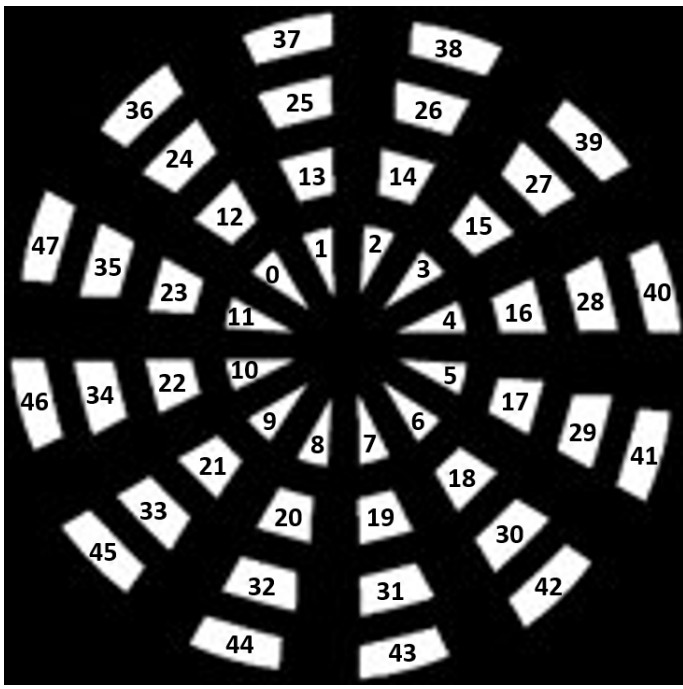

Positions of the pixels and their corresponding indices.

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
