# Peer review of "Quantum Wavefront Shaping with a 48-element Programmable Phase Plate for Electrons"

_SciPost Physics, doi:SciPost Phys. 15, 223 (2023)_

## Round 2 · Referee Report · Anonymous · 2023-10-31

Strengths
1) The authors present an interesting study on a novel phase plate for electron beams, with superior performance with respect to previously developed phase plates, including those developed by the authors themselves.
2) The phase sensitivity and phase resolution of the novel phase plate are investigated and reported. The advance with existing phase plate approaches is detailed in the introduction; notably that the phase plate can be altered with voltages in the mV range is a major advantage.
3) They also convincingly demonstrate the capabilities of the phase plate. In particular the experimental result on the ramped phase setting (Fig. 3(f)) for creating a vortex beam is impressive, but also the resolution improvement obtained by phase modification as described in section 3.3 is a straightforward proof of a useful application.
4) Future avenues are highlighted and the study is well connected to existing literature.
Weaknesses
None. It is a very strong and straightforward paper.
Report
The manuscript is of outstanding scientific quality and sufficiently novel. The manuscript is somewhat technological in nature, but based on advanced physical concepts. It also fits well into the scope of SciPost Physics, and can be accepted for publication as is.
Requested changes
I have only a few TYPO-level comments for the authors to consider:
p. 1, Abstract, line 7: microscopes are typically operated at a voltage, which implies kV, not keV. It is best to use keV only when it refers to the kinetic energy of the accelerated electrons. Please correct or rephrase.
p. 9, Figure 5, caption: “…for better presentation. and that the …”
p. 10, line 6: common aberration -> common aberrations
p. 10, line 12: Titan operation at 300 keV -> kV
p. 10, line -4: khz -> kHz

---

## Editorial Decision

published